# Critical Facets of European Corn Borer Adult Movement Ecology Relevant to Mitigating Field Resistance to Bt-Corn

**DOI:** 10.3390/insects15030160

**Published:** 2024-02-27

**Authors:** Thomas W. Sappington

**Affiliations:** 1Corn Insects and Crop Genetics Research Unit, Agricultural Research Service, United States Department of Agriculture, Ames, IA 50011, USA; tom.sappington@usda.gov; Tel.: +1-515-230-1441; 2Department of Plant Pathology, Entomology and Microbiology, Iowa State University, Ames, IA 50011, USA

**Keywords:** *Ostrinia nubilalis*, resistance, insect resistance management, dispersal, partial migration, ranging, station-keeping, flight, behavior

## Abstract

**Simple Summary:**

The European corn borer (*Ostrinia nubilalis*) is a major pest of corn, managed successfully in North America since 1996 with transgenic Bt-corn. However, practical field resistance and early-warning increases in frequency of resistance genes for all available insecticidal Bt proteins expressed in corn were detected recently at several locations in Canada. Given the high mobility of moths, design of containment and mitigation measures would have to be implemented at large spatial scales around a resistance hotspot to be successful. Effective strategies must account for high gene flow consistent with improved understandings of this species’ complex movement ecology, including migratory behavior.

**Abstract:**

The European corn borer (*Ostrinia nubilalis*, Hübner) has been managed successfully in North America since 1996 with transgenic Bt-corn. However, field-evolved resistance to all four available insecticidal Bt proteins has been detected in four provinces of Canada since 2018. Evidence suggests resistance may be spreading and evolving independently in scattered hotspots. Evolution and spread of resistance are functions of gene flow, and therefore dispersal, so design of effective resistance management and mitigation plans must take insect movement into account. Recent advances in characterizing European corn borer movement ecology have revealed a number of surprises, chief among them that a large percentage of adults disperse from the natal field via true migratory behavior, most before mating. This undermines a number of common key assumptions about adult behavior, patterns of movement, and gene flow, and stresses the need to reassess how ecological data are interpreted and how movement in models should be parameterized. While many questions remain concerning adult European corn borer movement ecology, the information currently available is coherent enough to construct a generalized framework useful for estimating the spatial scale required to implement possible Bt-resistance prevention, remediation, and mitigation strategies, and to assess their realistic chances of success.

## 1. Introduction

The European corn borer (*Ostrinia nubilalis* Hübner; Lepidoptera: Crambidae) is a serious pest of corn (*Zea mays* [L.]), introduced to North America from Europe in the early twentieth century [1]. It spread from initial introduction sites in Massachusetts, New York, and Ontario southward to Florida and westward to the foot of the Rocky Mountains by the late 1970s [2,3]. It is a polyphagous insect attacking a wide variety of plants [4], including crops like potato (*Solanum tuberosum* L.), green beans (*Phaseolus vulgaris* L.), and peppers (*Capsicum annuum* L.) [3,5,6,7,8,9,10]. Corn is the preferred host under most environmental conditions, however, and is the greatest contributor to European corn borer populations in the landscape [11,12,13]. Egg masses are usually oviposited on the underside of a leaf. Early instars (1st–3rd) feed on leaf tissue and larger instars (3rd–5th) tunnel inside the stalk or ear shank causing damage by disrupting water and nutrient flow, as well as structural injury and lodging which hinders harvest [14,15]. Timing of insecticide sprays is critical to ensure contact with as many larvae as possible before they can shelter inside the stalk or ear [16]. Fifth instars either pupate inside the stalk, or they enter diapause to overwinter, followed by pupation and emergence as adults the next spring or early summer.

Adults are nocturnal and live about 10–15 days [3]. They spend the day and much of the night in dense vegetation, which provides a humid microclimate. In the rainfed parts of the Corn Belt and Europe, they concentrate in grassy road ditches, fence lines, and waterways outside the field, unless the cornfield harbors thick patches of weeds; their distribution in the grass is contagious, creating aggregation (or “action”) sites [17,18,19,20,21,22,23,24,25,26,27,28]. Although some mating occurs in the cornfield [27], most occurs in the grassy aggregation sites. In semiarid regions like the Great Plains, where most fields are pivot-irrigated, lush grassy habitat outside the field is sparse and the adults often reside in the cornfield itself, which represents the most humid habitat in the landscape [29,30,31,32].

### 1.1. European Corn Borer Voltinism and Pheromone Races

The number of generations per year depends partly on the voltinism race of a European corn borer population. Diapause is obligate in the univoltine race. The bivoltine race usually has two generations in most of the U.S. Corn Belt; but diapause is facultative depending on the interaction of photoperiod and temperature, and can range from one to four generations per year contingent on latitude and environmental conditions [3,33]. In areas of overlap, the two races are partially reproductively isolated because of temporal differences in adult activity during the season [34,35,36,37]. For example, bivoltine-race adults in the Corn Belt are often active for about three weeks in late May through early June, and during a second “brood” or “flight” from late July through mid-August; univoltine adults have a single period of activity mainly in July [3,35,38,39]. Superimposed on voltinism races are two pheromone races designated “E” or “Z”, reflecting predominance of (*E*)- or (*Z*)-11-tetradecenyl-acetate in the blend of the two isomers comprising the female sex pheromone [40,41]. These two races overlap in parts of eastern North America, where they hybridize to some extent [13,35,42,43,44]. Combinations of voltinism and pheromone systems result in three races of European corn borer [45]: univoltine-Z (UZ), bivoltine-Z (BZ), and bivoltine-E (BE); there is no univoltine-E race. Gene flow among the races is mainly confined to autosomal genes, with race integrity maintained in part by co-adapted gene complexes controlling reproductive isolating mechanisms located on the sex chromosome [43,46,47,48,49].

### 1.2. Pest Management

Transgenic corn expressing one or more insecticidal proteins derived from the soil bacterium *Bacillus thuringiensis* (Bt) primarily targeting European corn borer was first commercialized in 1996 in the U.S. and 1997 in Canada [50], and was rapidly and widely adopted by farmers [51]. The first commercialized product expressed the Bt protein Cry1Ab, followed in later years by those expressing Cry1Fa and Cry1A.105 + Cry2Ab, and various combinations of these three in the same plant as pyramids. Control of this pest by Bt-corn hybrids has been excellent, providing dramatic area-wide population suppression, to the benefit even of those planting nonBt-corn or vegetables in the same landscapes [10,52,53,54]. Before the advent of Bt-corn, European corn borer was a chronic pest in agricultural landscapes where corn was grown, often present in large numbers. Because of the difficulty of properly timing insecticide treatments, many growers of field corn simply accepted unknown, but often significant, annual yield losses [3,15]. Despite the almost total protection Bt-corn provides against European corn borer, it remains the major pest of concern for many corn producers, perhaps because those farming during and before the early 1990s remember the losses it inflicted [55].

An ongoing concern has been loss of Bt-corn hybrid efficacy via evolution of resistance by the insect to one or more of the Bt Cry proteins [56,57]. The U.S. Environmental Protection Agency (USEPA) and Canadian Food Inspection Agency (CFIA) recognized the improved environmental safety Bt-corn affords to management of European corn borer over chemical insecticides and thus its value to society. In consequence, both agencies require that insect resistance management (IRM) strategies be implemented to prevent or delay evolution of resistance and loss of Bt-corn products [58,59,60].

### 1.3. Insect Resistance Management

IRM requirements for Bt-corn in both the U.S. and Canada are based on the high-dose refuge (HDR) strategy [60,61,62,63,64,65,66,67]. A “structured” (or “block”) refuge of nonBt-corn is planted nearby a field of Bt-corn, or in circumstances where it is allowed the nonBt refuge seed is mixed with Bt-corn seed for planting (“refuge-in-a-bag”, RIB) [60,67,68,69]. The refuge serves as a nursery of susceptible insects to mate with any rare resistant individuals emerging from the Bt-corn. A product is considered high dose if it kills all or almost all heterozygous-resistant individuals [60,70,71]. Because the high dose of the toxin renders resistance functionally recessive, offspring of such a mating will die on Bt-corn, purging the resistance allele each carries from the population [27,61,72,73]. In addition, the companies that developed the Bt seed are required to monitor for resistance annually by assaying populations for changes in susceptibility in geographically representative areas of high Bt-corn adoption; the goal is to provide early warning of increasing resistance [56,73,74]. “Early warning of resistance” is a classification that denotes a significant decrease in susceptibility to a Bt toxin but not at a level affecting crop efficacy against the insect in the field. “Practical resistance” is attained when >50% of the individuals in a population are resistant and efficacy of the crop has been reduced; both types of resistance are genetically-based and field-evolved in response to selection [67,71,75,76,77].

Other factors that delay evolution of resistance include associated fitness costs and low frequency of resistance alleles [63,71,78,79,80]. Fitness costs of Bt resistance in European corn borer appear to be weak [81,82]. A pyramid of two or more toxins in the same plant targeting the pest can also delay resistance as long as each is independently high-dose (causing so-called “redundant killing”) and if there is no cross-resistance between the toxins [69,83,84,85]. Early IRM for Bt-corn targeting European corn borer required at least 20% of a farmer’s corn to be structured refuge planted within 0.8 km of the Bt field [60,74]. In the case of pyramided products, a 5% RIB is now allowed instead [74], which is logistically more convenient for farmers compared to planting a separate block refuge, and ensures compliance with the IRM refuge requirement while allowing the farmer to plant a higher percentage of Bt-corn [68,69]. However, RIB is of uneven value in slowing resistance evolution depending on the target species’ movement ecology, and its use can be problematic with a mobile species, like European corn borer, that disperses from the natal field before mating [60,67,86].

European corn borer and the Bt-corn hybrids targeting them meet many of the criteria ideal for the HDR strategy to be effective. Cry1Ab, Cry1Fa, and Cry1A.105 + Cry2Ab hybrids are all high-dose against European corn borer [60,87]. Initial frequency of any resistance allele of a major gene in populations tested was very low for Cry1Ab, and all indications are that Cry1Ab resistance is functionally recessive [60,71,82,88]. In contrast, Cry1Fa resistance appears to be controlled by a major gene, and initial frequency of resistance alleles indicated they are not rare [82]. High mobility of this species [89] probably ensures most resistant adults emerging in a Bt field encounter and mate with susceptible adults emerging in a refuge block within the prescribed 0.8 km distance, and most likely well beyond that if refuge placement is not perfect.

## 2. Field-Evolved Resistance to Bt-Corn in Canada

IRM based on the HDR strategy for European corn borer in Bt-corn has been quite successful, with no reports of practical field resistance during the first two decades after its commercialization [57,66,71,76]. Bt-crops targeting other pests have not always fared so well, with steadily growing reports of confirmed practical field resistance [67,71]. Unfortunately, field-evolved practical resistance to Cry1Fa corn has now been documented in European corn borer populations at several locations in Nova Scotia, Quebec, and Manitoba in Canada [50,87]. The genetic basis of this resistance has been linked to mutations in the *ABCC2* gene, which codes for a receptor that binds Cry1Fa, and potentially contributes to cross-resistance with Cry1Ab and Cry1A.105 [90]. In addition, early-warning levels of field-evolved resistance to Cry1Ab have been detected in Nova Scotia and Manitoba; to Cry1A.105 in Nova Scotia, Prince Edward Island, and Quebec; and to Cry2Ab in Nova Scotia and Quebec [87].

Both de novo evolution of Bt-resistance in a European corn borer population and its spread are functions of gene flow, and therefore of adult movement [28]. Resistance in a population evolves based on the interplay of gene flow and selection for reduced susceptibility. Susceptible immigrants arriving to mate with resistant adults in the focal population will reduce the rate of resistance evolution in that population, which is the underlying rationale for the refuge strategy. In the absence of gene flow, evolution of local adaptations like Bt-resistance can be rapid in small, isolated populations [91,92,93], especially if nonBt-corn refuge is not planted as prescribed and Bt-corn traits are deployed singly rather than pyramided with another toxin. These factors seem to have combined in the small, isolated corn fields of Nova Scotia, leading to the first instances of sustained practical resistance of European corn borer to Bt-corn [50,87]. Annual sampling of European corn borer for Bt-resistance monitoring in the U.S. has focused solely in three regions of the Corn Belt where selection was expected to be greatest [56,74]. However, Kim et al. [94,95] cautioned that resistance monitoring should also be conducted in the northeastern U.S., where corn hectarage is relatively low, agricultural landscapes are diverse and fragmented, and potential topographical barriers to gene flow like forests and mountain ranges are common, creating relatively small, isolated populations. Previous monitoring likewise had not been conducted in the Canadian Maritime Provinces [50,87], where relatively small cornfields may be isolated by forested areas and hills.

The rate of resistance spread, measured either as an increase in resistance allele frequency or frequency of resistant individuals in the receiving population (see [73]), depends on the rate and distance of adult dispersal from the source location. An overall increase in geographic extent of resistance can also result from independent evolution of resistance in localized hotspots [86]. It is possible that European corn borer resistance to Cry1Fa Bt-corn in Canada is spreading from Nova Scotia, although similarities in agricultural landscapes in northeastern North America where corn is not a dominant component [96] suggest evolution creating independent hotspots is at least as likely. Independent evolution of resistance is certainly suspected in the Manitoba population, given its distance from Quebec and the Maritime Provinces [87]. Quick response is needed after detection of an early-warning level of resistance, such as that to Cry1Ab, Cry1A.105, and Cry2Ab2 in Canada, if remediation (reduction of resistance allele frequency) or mitigation (containment or slowing the spread of resistance alleles) is to be successful [73,74]. The imperative for quick action is even greater if trying to mitigate spread of resistance once a population of a mobile species like European corn borer has reached the stage of practical resistance [74]. The substantial spatial scale of detected Cry1Fa practical resistance within and beyond the Maritime Provinces [87] illustrates this point, and it portends the magnified challenge of containing it now that its presence is established far from the boundaries of the first-detected, localized hotspot.

## 3. Adult European Corn Borer Movement and Dispersal

A thorough understanding of adult European corn borer movement ecology is therefore of critical importance in designing and implementing feasible measures to remediate or at least mitigate the practical resistance to Cry1Fa Bt-corn documented in eastern Canada and the early warning of resistance to all Cry toxins before it spreads from existing hotspots or evolves in new ones. The purpose of this paper is not to propose mitigation strategies, but to review what is known about adult European corn borer flight behavior and its relation to mating behavior and gene flow, which must be considered when designing and implementing such strategies [97]. Questions about European corn borer movement ecology of particular interest in this context include: How far do adults (genes) travel per generation? What proportion of a population disperses different distances? When does mating and oviposition occur relative to when dispersal occurs? And where, spatially in the landscape, do mating and oviposition occur relative to the natal field?

Despite more than a century of research in North America, Europe, and Asia on this insect’s ecology and biology, and hundreds of papers published, a coherent holistic understanding of adult European corn borer movement ecology has been difficult to establish. Studies using different methods—e.g., rates of range expansion, mark-release-recapture (MRR), flight mills, observation and sampling, and population genetics—bearing on the general question “how far do adults disperse?” often produce apparently contradictory results. The studies themselves and the data generated are solid, so the differing results do not arise from flaws in methodology. The difficulty is in interpretation of the data based on an assortment of assumptions about adult behavior that may or may not be correct under the conditions of the study or when extrapolated to other conditions.

In coming to grips with this issue of putting together a coherent picture for European corn borer movement, it is important to keep in mind several principles: (1) Dispersal is defined in this paper as the geographic distance between parent and offspring [98]. Dispersal itself is not a behavior, but a spatial consequence of one or more types of movement behavior. (2) Dispersal is not a uniform event that can be characterized with a snapshot in time and then attributed as applicable throughout an adult’s lifespan under all conditions. (3) There is more than one type of movement behavior in the adult European corn borer repertoire that can result in spatial displacement. (4) The kind of movement behavior employed at any given time depends on an individual’s motivation to move, which can change from moment to moment, with age, mating status, and a multitude of other variables. (5) The movement path and net dispersal over a given time interval via a particular type of movement behavior under a given motivation is influenced by many interacting factors, both biotic and abiotic, all of which can vary in their own right [99]. (6) A dispersal event has different consequences for IRM depending on its spatial scale, when it occurs during the lifetime of an individual, and in relation to the movement and dispersal of other individuals in the population of interest. In other words, an individual’s movement behavior not only has consequences for that individual (displacement), but also has consequences at the population level. (7) Population-level consequences are an emergent property of movement by individuals in that population [100,101,102]. European corn borer movement ecology is complex, but recognizing and embracing the complexity paradoxically allows for the simplifications needed to construct a generalized framework useful for designing successful resistance prevention, remediation, and mitigation strategies. While there are still a number of mysteries related to adult movement, we do know enough to create such a framework.

### 3.1. Types of Movement

Foundational to characterizing the movement ecology of European corn borer adults is understanding the basic types of movement they may engage in. The categories below and in Figure 1 are described by Dingle and Drake [100], and Dingle [101]. At its broadest, most fundamental level, the motivation behind a given bout of movement is either *appetitive* or *nonappetitive*. Most lifetime movement is appetitive, during which an individual is seeking or searching for a resource such as a mate, oviposition site, sheltering habitat for daytime inactivity, escape from a pursuing predator like bats [103], etc. Appetitive behavior is facultative, triggered by proximate conditions. The movement path [99] during appetitive flight is generally meandering, with frequent turns and variable flight speeds. When the insect encounters the sought-after resource, its searching behavior is arrested and behaviors associated with use or engagement with the resource ensue. Because appetitive flight is not straight-line, but instead involves frequent turning and counterturning, the net displacement distance from the starting point to the resource will be less, often much less, than the total distance flown [104]. In addition, appetitive flight usually takes place within the flight boundary layer of the atmosphere near the Earth’s surface where windspeeds are lower than the insect’s flight speed, allowing the insect to control its flight direction as it searches its surroundings.

#### 3.1.1. Appetitive Flight: Station-Keeping and Ranging

Appetitive flight behaviors fall into two main types, *station-keeping* and *ranging* (Figure 1). Station-keeping movement is related to routine, maintenance activities that occur in the same relatively small area from one day to the next. This small area (“station”) constitutes the individual’s *home range*. Much of a European corn borer’s station-keeping movement is related to foraging for resources. For example, both sexes leave their grassy aggregation sites soon after sunset to search for water droplets from which to drink, flying over short grass where dew forms first [20]. Longevity and lifetime fecundity are reduced in laboratory females without access to free water [105], and the search for water droplets seems to take priority over other nightly activities. Once the imperative to imbibe water has been satisfied, reproductive activity proceeds. Males cast about over grassy habitat, or possibly within the grass canopy [16], throughout the night searching for a female pheromone plume; once detected, the male follows the plume in a typical zigzag flight pattern to its source [106,107]. Mated females fly into a nearby cornfield to oviposit. Unmated females return to the grass after searching for dew to release pheromone and mate [18,19,22]. Most mated females return to a grassy site after oviposition, but not necessarily the same ones they vacated earlier [26,27,28,108]. Moving daily between aggregation sites and the cornfield is indicative of a “multihabitat population” [109], and is a specialized form of station-keeping behavior called *commuting* [100,101].

Ranging is fundamentally very similar to station-keeping in that it is also appetitive behavior, but differs in that it permanently displaces the insect beyond its home range [110] (Figure 1). Ranging occurs when a sought-after resource is not detected nearby, leading to extended foraging in the landscape. Because it is appetitive behavior, ranging is arrested once the resource is encountered [110], and station-keeping behavior is resumed in what will now become the insect’s new home range. Ranging by European corn borer is suggested by aggregation habitat and host plant preferences expressed by females in a heterogenous landscape. Suitable grassy aggregation habitat is defined by proper microclimate, plant density, plant structure, and species composition [20,25], as well as proximity to a cornfield of an acceptable phenological stage for oviposition [26]. Such conditions may be plentiful in some landscapes, but may require a longer search in areas such as the semiarid Great Plains, or in regions outside the Corn Belt where corn is not the dominant component of a diversified or fragmented agricultural landscape, such as in the northeastern U.S. or the Maritime Provinces of Canada. Ovipositing European corn borer females show strong preferences for corn at certain phenological stages. At a given moment in a landscape containing cornfields of contrasting planting dates, some fields will be more attractive for oviposition than others [111,112,113]. Furthermore, some alternative host crop species are more attractive than corn if the latter is either too young or senescent when females are searching for an oviposition host [5,114,115]. In areas where Bt-corn has suppressed the overall European corn borer population [10,52], locating aggregations of adults in grassy roadside ditches is often difficult (author, personal observation), and it seems likely that the few males in the landscape may range over considerable distances, perhaps several km, before encountering pheromone plumes released by equally rare females.

#### 3.1.2. Nonappetitive Flight: Migratory

All European corn borer adults engage in appetitive flight behavior, probably every night of their lives. A portion of these same adults also engage in nonappetitive *migratory* behavior (Figure 1) [89]. Like ranging, migratory behavior displaces the individual permanently out of its home range. But migratory flight is fundamentally different in that it is straight-line, undistracted by encounters with resources (such as habitat, food, or potential mates), is pre-programmed rather than triggered by proximate environmental conditions, and is often wind-aided at elevations above the flight boundary layer [100,116,117,118,119]. European corn borer is not typically thought of as a species that migrates, because its migration is aseasonal, undirected, and occurs within a permanent year-round distribution [89]. It overwinters in-place in diapause rather than moving between a separate overwintering habitat in the south and reproductive habitat in the north. Thus, European corn borer migration is not characterized by the mass springtime exodus from overwintering grounds and mass arrival in a disjunct breeding range that makes a migration event so obvious for seasonal migrants like black cutworm (*Agrotis ipsilon*) [120], fall armyworm (*Spodoptera frugiperda*) [121,122], monarch butterfly (*Danaus plexippus*) [123,124], and many others [125]. Instead, migration is almost invisible for an insect like European corn borer, which departs and arrives where others of its kind are simultaneously arriving and departing [89,126,127]. There is no spatial goal to which they orient, so the direction the moths travel is dependent on wind direction on the night of ascent into the atmosphere. Migration is a process defined by behavior, not by the consequences of that behavior [100,101,119,128,129,130,131]. The population-level consequence of migration in European corn borer is not a mass displacement over latitudes, but a spatial reshuffling or mixing of individuals within a stable species-level spatial distribution.

### 3.2. Evidence for Migratory Flight by European Corn Borer

Migration is not defined by absolute distance traveled, which can be short or long or a continuum depending on the species [101,110,129]. However, because migration is a process defined by straight, persistent, undistracted flight behavior, net displacement of a migrating insect will be greater per unit time than achievable by meandering appetitive flight. Thus, displacement of an insect over long distances relative to those typically covered by conspecifics during station-keeping or ranging flights is itself evidence for migratory flight. The distribution of lifetime dispersal distances of an organism engaged only in appetitive behaviors is assumed to result in a population-level dispersal pattern of *diffusion*. The resulting spatial pattern of diffusion for a cohort of adults emerging from the same natal location is generally envisioned as a two-dimensional Gaussian (normal) distribution of dispersal distances from the point of origin [98,132,133]. In contrast, migratory flight usually displaces an insect over longer distances than achievable by meandering appetitive flight, resulting in movement by *leaps*. The evidence for migratory behavior in European corn borer is manifold, as reviewed below and in detail elsewhere [89].

#### 3.2.1. Range Expansion and Capture after Crossing Extensive Barriers

Migratory leaps by European corn borer are most evident in range expansion data, because colonization of corn in an area beyond an invasion front is more easily detectable than when migration occurs within its year-round distribution. This species’ range expansion through North America averaged 46–57 km/generation [126,134,135] for the BZ race. Average rates for the UZ race were lower in North America (19 km/gen; [135]) and more recently in Germany (10–15 km/gen; [136,137]). Expansion along the main invasion front is typically by diffusion, but leaps ahead of the front indicate stratified dispersal [138,139], suggesting part of the population is migratory. For example, range expansion of the UZ race into Ohio via probable wind-aided flight of 45–80 km across Lake Erie from Ontario [1] represented a leap ahead of the invasion front, and leaps of at least 40 km/gen occurred in Germany as the UZ race expanded its range northward [137]. Leaps of about 53 km/generation in South Dakota and 74 km/generation in Minnesota during invasion by the BZ race are inferred from annual larval population surveys along east to west and southeast to northwest transects, respectively [126]. Dispersal distances based on leaps ahead of an invasion front are minimum estimates of dispersal capacity because it usually takes more than one mated female to successfully colonize a new area [140,141], and individuals that disperse farther without establishing a new population will not be detected. At the same time, the numbers in a founding population are normally small and it may take more than one year for the new population to be detected [142]. Meanwhile the population front draws nearer, decreasing the estimate of minimum distance that must have been traversed by colonizers. Similarly, the seeding individuals could have emigrated from somewhere behind the advancing front or some other more distant location [143]. Long-distance colonization events made evident during range expansion is indicative of nonappetitive migratory behavior because intervening areas of abundant suitable habitat did not arrest flight, as would be expected if the insects were searching for habitat via appetitive ranging.

Long-distance leaps are also inferred when individuals are captured in traps outside their normal range and far from the nearest likely source [144]. For example, European corn borer were captured in Finland [145] and the U.K. [146,147,148], where the nearest possible sources required flight across wide expanses of water. Although theoretically the individuals could have been engaging in ranging flight, they were captured in the company of other well-known migratory insects arriving on southerly winds. Likewise, a European corn borer moth was captured in Israel in the company of migratory beet armyworm (*Spodoptera exigua*) at least 150 km across uninhabitable desert from the nearest possible source area [149].

#### 3.2.2. Mark-Release-Recapture

To my knowledge, eight mark-release-recapture (MRR) studies of European corn borer adults have been published [1,27,28,30,31,108,150,151], all but one of which included replicated trials (see Table 1 in [89] for summary). Like any methodology for studying insect movement, MRR studies have strengths and weaknesses. The major strength of the MRR strategy is that a recapture definitively documents movement of an insect from the release site to the point of recovery. A well-known weakness is the dilution effect on recapture efficiency as the space between dispersing insects increases with distance from the release point, substantially limiting the spatial scale of sampling that is practical [104,110,152,153]. This means that lack of recapture at any distance cannot be construed as lack of flight to that distance. Several of the European corn borer MRR studies recorded maximum flight distances of <1 km [27,28,30,31,151], but this was a function of recapture arenas that did not extend beyond a few tens or hundreds of meters. Nevertheless, two studies recaptured marked moths at substantial distances from the release site. Caffrey and Worthley [1] released >60,000 marked European corn borer in the daytime on an ocean beach in Massachussetts and recaptured one male 32 km across Cape Cod Bay, flying with the wind. They released several thousand more on the beach and recaptured four females and three males after they flew 8 km along the beach with the wind. Because their experiments were conducted under environmental conditions (daytime release on unvegetated beaches) grossly different than those encountered by moths in wild populations, it is unlikely the flight behavior of the released moths was migratory, but the results do indicate the capacity for wind-aided flight over those distances. Showers et al. [150] released a total of >600,000 internally marked European corn borers the night after emergence over three years in six release-date windows of 3–5 separate releases each. Males were targeted for recapture by pheromone traps deployed in square rings out to a maximum distance of 75 km. Three marked females were recovered fortuitously during the course of the experiments. Male recapture rates ranged from 0.12–0.33% over the course of the study, most within 3.2 km, but trap density decreased with distance. Eleven marked moths were recaptured at distances >7 km, with three recovered at >40 km, including a marked female accidently captured in a pheromone trap at 49 km from the release site. Use of pheromone traps or light traps for recovery in MRR studies of European corn borer [30,31,150] introduces another set of potential problems in interpretation of captures or non-captures, related to the traps’ reliance on successful manipulation of the insect’s behavior. Pheromone traps are especially problematic in that recovery data are restricted to males, the male must be sexually responsive during the trial, and capture efficiency may be affected by competition from feral females [154,155] and trap placement [16,156,157]. Nevertheless, maximum recapture distance represents the minimum distance the insect is capable of moving.

Just as a sudden appearance of large numbers of adults at a location can indicate an immigration event, a sudden disappearance from a location is evidence of emigration. Both kinds of events may go unnoticed if they occur via migratory behavior within a year-round distribution. While MRR studies cannot generally provide a clear picture of typical flight distances of released moths for the reasons outlined above, they can provide good evidence for mass departure from a release site when the experiment is designed to measure it. This is the case for most of the European corn borer MRR studies. Investigators in these studies generally assumed the released moths would occupy grassy field borders (or irrigated corn in arid regions) at or near the release site, but nearly all reported very low mean percentages of nearby recovery, usually <5% and often <1%. The authors of all the European corn borer MRR studies concluded that most released moths must have flown beyond the furthest extent of recapture attempts based on evidence specific to their experimental setups, including spatial and temporal distributions of the recaptures, and comparisons with those of feral moths and their presumed source fields in the landscape [27,28,30,31]. In most studies, the number of moths that left the release site was either specifically known [1,27,28] or was calculated by subsampling adult emergence from corrugated cardboard rings of laboratory-reared internally-marked pupae [31,108,150,151]. Sampling in or near the release site was conducted in grass by the flush bar technique [150], light traps [30,31], pheromone traps [31], or sweep nets [1,27,28,108,151]. In the experiments reported by Dalecky et al. [27] and Bailey et al. [28], sweep netting was exhaustive in and near the grassy release sites, resulting in near 100% capture of any moths present. In all the MRR studies (except [1]), the presence of feral moths in nearby sampled habitat provided a positive control for habitat suitability. That acceptable habitat near the release sites did not arrest flight activity of most released moths, a signature of nonappetitive behavior, suggests the departures were via migratory flight rather than via appetitive ranging. It is possible the high density of moths at a release site induced emigration by flight akin to escape behavior, which is appetitive. The releases in several studies [30,31,102,146] were of internally-marked pupae, from which adults were allowed to eclose in the field, thus reducing possible effects of handling on flight behavior. Although density-stimulated departure cannot be ruled out without additional experimentation, there are other lines of evidence supporting a high rate of emigration by adults from the natal field before reproduction.

#### 3.2.3. Flight Mill

The frequency distribution of European corn borer flight duration and distance on rotary flight mills showed most are short, falling away steadily to more uncommon long flights, some lasting the full 8 h of darkness in the flight chamber [97,158]. The median duration of the longest uninterrupted flights of females was greatest (>2 h) on the night after emergence (1-day-old), but most 2- and 3-day-old adults of both sexes also made a long flight of >1 h duration [97]. In many insect species, migratory activity is limited to the preoviposition period, a common component of a species’ migratory syndrome consisting of various physiological, developmental, and behavioral mechanisms supporting the process of migration [100,118]. The preoviposition period of European corn borer is 3–5 d [159], so longest duration of uninterrupted flight occurring on days 1–3 after adult eclosion is consistent with this common attribute of insect migratory syndromes.

Maximum distances on the flight mill during a continuous flight reached >20 km [97]. Median flight speeds of 1-day-old females and males on the mills (0.52 m/s and 0.26 m/s, respectively [97]) were much slower than that of two marked males and a marked female (all 1-day-old) recovered 100 min after release flying together 14 km downwind [150]. Subtracting speed of the tailwind indicated a free-flight speed of 1.47 m/s, 3–5X greater than on the flight mill, reflecting drag at the mill pivot and weight of the flight arm. Wind-aided flight of any insect can be much faster still [100,119,125,160], emphasizing that flight distances of >20 km observed on flight mills must be considered minimum estimates of distance attainable by European corn borer in nature.

#### 3.2.4. Population Genetics and Gene Flow

Allele frequencies at selectively neutral genetic loci change over time in a population by chance (i.e., by drift) at a rate largely determined by effective population size. The rate of genetic differentiation of two populations separated in space is a function of the interplay between genetic drift and the rate of gene exchange between the two populations [98]. Such gene exchange, or gene flow, is dependent on movement of individuals between the populations (called “migration” in the population genetics literature regardless of the movement behaviors involved) followed by reproduction. The higher the rate of gene flow (effective migration) between populations per generation, the slower the rate of differentiation. The most common measure of genetic differentiation between populations at a point in time and across a panel of genetic markers is *F*_ST_ [98,161,162], which can provide insight into average historic rates of effective migrant exchange, and hence distances and patterns of dispersal [98,163,164,165,166]. *F*_ST_ values can range from 0 to 1, with low values across genetic markers indicating low differentiation and potentially high gene flow, and high values indicating high differentiation and potentially low gene flow [162,167,168]. In a case where the metapopulation essentially occupies a range of continuous reproductive habitat, such as with European corn borer in the U.S. Corn Belt, genetic differentiation increases with distance. This relationship of isolation-by-distance (IBD) in a two-dimensional habitat is detected by regression of genetic distance (i.e., *F*_ST_/(1 − *F*_ST_)) on the logarithm of geographic distance. A shallow slope suggests effective dispersal across greater geographic distances than does a steep slope.

At least 13 population genetics studies of European corn borer in North America and Eurasia have been published, reporting pairwise *F*_ST_ (or similar parameter) estimates between geographic locations using a variety of genetic marker systems (Table 1). *F*_ST_ estimates are consistently low and seldom statistically significant, suggesting historic gene flow over long distances [94,169,170,171], including estimates of 240–720 km in the U.S. [94] and ~600 km in France [172,173]. Estimates of IBD were usually nonsignificant, the only exceptions being when maximum pairwise geographic distances were >500 km [95,172,174]. Pairwise comparisons of differentiation between nearby sites likewise indicated very low and nonsignificant *F*_ST_ values. These included six comparisons between adult samples taken at 16 km intervals along an east–west transect in central Iowa [94]; three comparisons across distances of 7.5 to 13 km in Pennsylvania [44]; and 9 of 10 comparisons between sites in southern France separated by 2.7 to 16 km (one *F*_ST_ of 0.013 at 12 km distance was significant) [173]. Most of the few significant *F*_ST_ values reported for pairwise comparisons were either over long distances or were associated with areas of diverse cropping systems and fragmented agricultural landscapes. Bourguet et al. [172] noted that the only region in France with significant isolation by distance was in the northwest, where cornfields were fewer and more scattered than in the south and northeast regions of the country. Temporal estimates of change in allelic variation [175] did not differ within locations at six sites over 4–6 generations in France [172], and temporal estimates of *F*_ST_ were not significant within four sites over 3 generations in Iowa [94], indicating high stability of allele frequencies over time (Table 1).

The distribution of lifetime displacement distance for a particular species can be calculated using population genetics data in combination with estimates of population density from field sampling, based on the concept of Wright’s genetic neighborhood [98,132,133]. The parents of ~87% of individuals are expected to have originated within the radius of the neighborhood area, which is generally assumed to be a function of diffusive movement. Thus, the radius is an estimate of the lifetime effective dispersal distance of ~87% of individuals in a population [163,176], and has been estimated as ~12 km for European corn borer [95]. Reciprocally, ~13% of adult European corn borer in a population are estimated to fly farther than 12 km in their lifetime before reproducing.

**Table 1 insects-15-00160-t001:** Summary of overall and pairwise *F*_ST_ (or similar parameter, e.g., θ, *G*_ST_) estimates between European corn borer populations reported in the literature.

		Pairwise Geographic Scale (km)							
Region ^a^	No. Sites	Min	Max	Genetic Marker System	Signif. IBD?	Overall *F*_ST_ or Pairwise *F*_ST_ Range ^b^	% Signif. Pairwise *F*_ST_’s	Notes	Citation
Eurasia:								Northern populations UZ, southern BZ. For temporal analyses: high stability of allele frequency distributions within each site over 4–6 generations.	Bourguet et al., 2000 [172]
France (Spatial)						(θ)			
Total	29	<1	863	Allozyme	Yes	0.011 *	0		
*Northeast*	11	<1	286		No	0.015 *			
*Northwest*	7	32	509		Yes	0.012 *			
*South*	11	<3	252		No	0.003 ^NS^			
(Temporal)									
*South*	6	0	64				0 (change in allele frequency)		
Eurasia:						(θ)			Bourguet et al., 2000 [177]
Northern France	3	19	44	Allozyme	No	0.012 *	—		
Eurasia:						(θ)		6 and 3 samples for allozymes from Bourguet et al. [172] & [177], respectively. mtDNA samples from Bourguet et al. [172].	Martel et al., 2003 [169]
Northern France	9	19	503	Allozyme	No	0.009 *	—		
	5			mtDNA haplotype	—	0.039 *	—		
North America:								14 populations including BZ, UZ, & BE races.	Coates et al., 2004 [178]
USA: 8 states, ME to KS	14	81	2639	mtDNA haplotype	No	0.024 *	10.4		
*BZ only*	10	81	2639		—	—	0		
Eurasia:						(θ)			Leniaud et al., 2006 [170]
France	9	47	500	Allozyme	No	0			
Eurasia:									Malausa et al., 2007 [173]
France	14	3	768	Microsat	No	0–0.019	34.6		
*Small scale, North*	7	3	16			0–0.014	10.0		
North America:								Some sites may have included both UZ and BZ races. One site was far south of Corn Belt in LA. Significance of *G*_ST_ values was not reported.	Krumm et al., 2008 [171]
USA: 11 states,						(*G*_ST_)			
western Corn Belt	18	44	1900	AFLP	No	0.115	—		
*North*	6	44	500		No	0.203			
*West*	6	108	303		No	0.171			
*South*	6	142	1028		No	0.153			
North America:								Spatial samples along 2 transects (N-S, E-W). Temporal comparisons over 3 generations. Temporal maximum likelihood estimates of migration rate ^c^ (*m*) ranged from 0.04–0.54; geographic pattern of *m* reflected prevailing summer wind direction.	Kim et al., 2009 [94]
Central Corn Belt									
USA: 5 states, MN, IA, MO, NE, IL, (Spatial)									
Total	13	16	720	Microsat	No	0–0.013	1.3		
*Small scale, along*	6	16	80		No	0–0.007	0		
*E-W transect, IA*									
IA (Temporal)									
Between sites	4	170	240		—	0–0.014	8.3		
Within sites	4	0	0		—	0.001–0.002	0		
North America:								Estimated Wright’s genetic neighborhood radius as ~12 km.	Kim et al., 2011 [95]
USA: 8 states, New York to Colorado	12	64	2180	Microsat	Yes	0–0.012	28.8		
Eurasia:									Frolov et al., 2012 [174]
Russia	6	71	679	Microsat	—	0–0.013	20.0		
Plus 7 France sites from [145]	13	71	2741		Yes	—	—		
North America:								Samples from N-S transect of [90], 80-km intervals.	Levy et al., 2015 [33]
Central Corn Belt									
USA: 3 states, MN, IA, MO	10	80	720	SNP	NO	0.002 ^NS^	—		
North America:									Coates et al., 2019 [44]
Northeastern USA: 2 states, NY, PA:									
BZ race	12	7.5	259	SNP	—	0–0.013	3.0		
*Small scale (≤16 km)*	3	7.5	13.4		—	0–0.001	0		
BE race	12	7.5	259		—	0–0.009	7.6		
*Small scale (≤16 km)*	3	7.5	13.4		—	0–0.001	0		
BZ + BE (combined per location)	12	7.5	259		No				
Eurasia:									Li and Yang 2022 [179]
Western China: Yili valley, Xinjiang	3	53	204	mtDNA haplotype	No	0–0.100	—		
	3	53	204	Nuclear haplotype	No	0–0.057	—		

IBD, isolation by distance; Microsat, microsatellite; AFLP, amplified fragment length polymorphism; SNP, single nucleotide polymorphism. USA state abbreviations: CO, Colorado; IA, Iowa; KS, Kansas; LA, Louisiana; ME, Maine; MN, Minnesota; MO, Missouri; NY, New York; PA, Pennsylvania. European corn borer voltinism-pheromone race designations: UZ, univoltine-Z; BZ, bivoltine-Z; BE, bivoltine-E. *, Significant (*p* < 0.05) overall *F*_ST_; ^NS^, non-significant overall *F*_ST_. ^a^ Regions in italics are subsets of parent region presented above it. ^b^ Negative *F*_ST_ estimates in original papers are reported here as ‘0’. ^c^ Migration rate (*m*) indicates predicted proportion of population consisting of immigrants from other three populations.

#### 3.2.5. Population Densities in a Cornfield Are Uncorrelated between Generations

The population density of European corn borer in a field during one generation is uncorrelated with the population density in the same field the next generation, whether within or between years [126,180,181,182,183]. For example, although crop rotation can provide some benefit against European corn borer, its influence is apparently indirect through enhancement of natural enemies [184] or possibly soil-health impacts on oviposition preference [185]. However, any such benefits are not of enough practical consequence to merit recommendation of crop rotation for protection from this pest [3,186]. Similarly, destruction or burial of cornstalk residue after harvest greatly reduces European corn borer adult emergence, but has no substantive effect on population infestation level in that field the following season [187,188]. For cultural controls like crop rotation or stalk destruction to suppress European corn borer infestations the following generation, they would have to be implemented on an areawide scale, analogous to the documented case of regional suppression by widespread Bt-corn adoption [10,52,54]. This generational independence of population levels in a given field is consistent with emigration of most emerged adults before reproduction and de novo infestation of each field each generation, predominantly by immigrants. Although a number of stochastic weather-related factors may reduce between-generation correlation of infestation levels by impacting mortality [180,181,189], a field-conducted life table study [190] showed loss of adults by emigration was by far the most important factor limiting population growth in an isolated cornfield over generations [191]. Flush sampling in the spring in central Iowa showed that, as expected, the current crop affected distribution in the local landscape of European corn borer adults in grassy field borders [26]. Unexpectedly, however, the presence of corn stubble from the previous year was unrelated to adult distribution in the landscape, again indicating newly emerged adults emigrated from the vicinity of their natal fields and infestation of all fields was predominantly by immigrants.

### 3.3. Proportion That Migrates

The 1–2 orders of magnitude difference in low dispersal distances estimated from direct experimentation (e.g., MRR, flight mills) and the long distances estimated indirectly from population genetics estimates of gene flow is a phenomenon called “Slatkin’s paradox”, which has been reported from several taxa [167,192,193]. A similar kind of phenomenon, called “Reid’s paradox” [193,194,195,196,197], involves cases where direct measures of movement capacity, again like MRR and flight mill studies, indicate much shorter dispersal distances than the long-distance estimates derived from other ecological data like rates of range expansion or detection of individuals far from the nearest potential source population. Neither Slatkin’s nor Reid’s paradox are indicators of flawed methodology, but most often arise from a bimodal distribution of lifetime dispersal distances among individuals within the species. This seems to be the case with European corn borer. It is most likely a *partial migratory species* [89], meaning that some portion of any population (*migrant* behavioral phenotype) migrates relatively long distances from the initial home range by leaps, while the other portion (*resident* behavioral phenotype) remains in its natal landscape, dispersing by diffusion via relatively short-distance appetitive flights [130,198,199,200]. The small percentage of marked European corn borer that remain nearby in MRR studies supports the conclusion that this is a partial migratory species. Partial migration is the most common type exhibited across migratory taxa [198,199]. It is an example of a mixed dispersal strategy based on maternal bet-hedging to optimize survival of offspring by balancing the potential costs of staying in the natal environment (e.g., inbreeding, build-up of disease) with those associated with emigration (e.g., exposure to hazards, possible absence of suitable habitat, suboptimal adaptation to new environment) [201].

The proportion of European corn borer adults that are of the migrant behavioral phenotype seems to be quite high compared to the proportion that are residents, although precise estimates are not yet possible, in part because the proportion may be naturally variable within and between populations (e.g., [27]). One cannot take the 12 km estimate [95] of lifetime dispersal of 87% of adults from Wright’s genetic neighborhood analysis as that which occurs only by diffusion via appetitive flight, and conclude therefore that 13% migrate. The normal distribution of Wright’s genetic neighborhood area is an a priori assumption of this method of calculation. It is more likely that the neighborhood area calculated for European corn borer derives from a conflation of two different dispersal kernels: one of short-distance appetitive flight by resident individuals which do not migrate, and the other of longer-distance movement by migrants (Figure 2). If so, the distribution of dispersal distances in a population would be fat-tailed or leptokurtic [195,196,202], with a long gradual decline, rather than a thin-tailed Gaussian distribution, with a sharp decline to very rare long-distance flights [202,203]. A fat-tailed distribution of dispersal distances would imply that the 12 km estimate for lifetime displacement is too high for diffusion by residents. Flight duration data from flight mills are consistent with such a fat-tailed distribution, where the number of European corn borer moths making flights of progressively longer duration only gradually declines, at least for females (Figure 3) [97,158].

Compared to distances flown by seasonal migrants like black cutworm or monarch butterflies, migrating European corn borer do not necessarily travel very far, with most of them probably settling within the ~12 km radius of Wright’s genetic neighborhood before conducting any necessary ranging to find suitable habitat. Nevertheless, several population genetics studies estimated migrant exchange among populations separated by hundreds of km [33,94,169,172] (Table 1), distances that are undoubtedly well beyond those attainable by meandering, appetitive ranging flights within the slow winds in the flight boundary layer where searching behavior takes place.

It is possible that migration in European corn borer is differential [100], in that females may travel farther distances than males on average, based on flight mill duration and distance data [97,158]. Median duration of the longest uninterrupted flight of females was highest on the first full night of emergence, whereas flight duration of males on the first night was significantly and substantially lower. On the other hand, MRR data show disappearance of males from the release area on the first night is as striking as that of females, suggesting that propensity to migrate is the same or similar. Thus, it seems likely that males do migrate, albeit with differences in developmental timing and distance.

### 3.4. Timing of Mating Relative to Timing of Dispersal

The relationship of timing of mating with that of dispersal from the natal field critically affects spatial patterns of gene flow and thus resistance evolution and spread [27,72,172]. It is clear that mating activity of European corn borer is strongly associated with aggregation sites in grassy habitat, usually outside the cornfield [18,19,20,22]. However, some mating occurred near or in the field of emergence in MRR experiments in France [27], and occurs within irrigated cornfields in semi-arid regions of the U.S. [29,30,31,32]. But in all of these environments, MRR studies also showed that most released adults departed the release area within a day (see Table 1 in [89]) probably via migratory behavior as discussed earlier. In laboratory experiments, most European corn borer females initiated calling (release of pheromone) on the first or second night after emergence, with none calling on the night of emergence [204]. This would seem to leave little time for pre-dispersal mating in or adjacent to the natal field if most adults emigrate by the first night after emergence. A total of 46 European corn borer females were individually observed in a cornfield from pupal eclosion to dawn during first and second flights in Minnesota; none released sex pheromone on the night of emergence, and all left the vicinity of the emergence corn plant by dawn [205]. However, in the Dalecky et al. [27] study, an average of 18% of females released when less than 24 h old remained in the vicinity of the release site long enough to mate locally. Patterns of long-distance flight behavior on flight mills did not differ between mated and unmated females of the same age, or between mated and unmated males of the same age [97]. Taken together with the rapid disappearance of marked moths from release sites in MRR studies, it is likely that both mated and unmated European corn borer moths engage in migratory flight, but that most (>80%) are unmated during migration.

## 4. Conceptual Framework of European Corn Borer Adult Movement Ecology

The balance of evidence available suggests the following conceptual framework of European corn borer adult movement ecology (Figure 4): A high percentage (probably >90%) of adults are of the migrant behavioral phenotype and emigrate from the natal field via nonappetitive migratory behavior before oviposition. Most migrants emigrate from the field on the first night after emergence but some migrate on the second or third night. Of the migrants, most (>80%) are unmated when they emigrate. Those migrants that mate locally before emigrating are most likely to have mated with immigrants which make up most of the adult population in adjacent grassy aggregation sites. A much smaller percentage of adults (variable, but probably <10%) mate locally and reproduce in the natal field. Most migration is terminated within 12 km of the natal field but ~13% of migrating adults travel farther, up to several hundred km away in wind-aided migration. Newly arrived, unmated migrants use ranging behavior to locate suitable habitat for mating, whether that be grassy aggregation sites adjacent to cornfields at a phenological stage suitable for oviposition [26] or within irrigated cornfields in semi-arid regions [30]. Further spatial mixing occurs nightly via commuting flights between grassy aggregation sites and the oviposition host, but these occur mostly within the new home range, with only short net displacement distances characteristic of meandering foraging and ranging flights. An important implication of this conceptual model of high pre-reproductive emigration rates via aseasonal, undirected migration is that a large majority of European corn borer adults observed occupying any grassy field border or ovipositing in any cornfield are immigrants, not local moths that emerged in that field or nearby. Experimental designs, interpretations of data, and IRM models for European corn borer have almost always assumed the opposite, namely that most of the moths sampled in or near a particular cornfield emerged, then mated, and are ovipositing in the natal field or nearby. This is almost certainly incorrect, and it matters because such interpretations influence assessment and design of IRM tactics and strategies.

## 5. Implications for Bt-Resistance Remediation and Mitigation

Given the high mobility of adults, remediation of European corn borer populations exhibiting practical Bt-resistance in identified hotspots will be very challenging. It may be possible in principle if the hotspot is small enough, say a single field or a small cluster of fields, by eliminating all or nearly all overwintering larvae, which should all be homozygous for the resistance allele, with stalk destruction and plowing before they emerge as adults and emigrate the next spring. Only the small percentage of residents emerging from that hotspot field after overwintering will remain to mate and oviposit, and most of those residents will mate with immigrants, which mostly originate from fields within 12 km (Figure 4). Thus, an important consideration will be the frequency of resistance alleles within 12 km of the hotspot. If a “containment zone” of some sort is possible, for example where stalk destruction and no planting of corn within a certain number of km is implemented, this will reduce the number of immigrants into the hotspot, and those that do arrive will be from farther away and presumably more likely to carry only Bt-susceptible alleles, helping to dissipate resistance.

The dimensions of a containment zone around a hotspot of resistance development is a critical question, often difficult to determine in practice [86]. In this case, an area of 12 km radius is supported by the estimate of typical adult lifetime dispersal for ~87% of European corn borer [95], but the radius may need to be even greater. O’Rourke et al. [96] found that European corn borer densities in nonBt cornfields in the agriculturally diverse region of upstate New York were unrelated to percentage of corn in the landscape at scales of the adjacent perimeter, 1-km, and 20-km radii. The authors attributed the lack of correlation to the high mobility and diet-breadth of this species, and suggested population density might be related to land use at a scale > 20 km. This result contrasted with that for western corn rootworm (*Diabrotica virgifera virgifera*, Coleoptera: Chrysomelidae), densities of which were significantly correlated with corn use in the landscape at 1-km and 20-km radii [96]. Western corn rootworm is also a partial migratory species, but with only about 25–30% of adults emigrating from the natal field before or in the midst of reproduction [127]. The 12 km radius, or whatever radius is chosen, may need to extend from the edge of the spatial extent of early warning resistance rather than from the area of observed practical resistance. Speed of implementation is critical for successful remediation, but determining the spatial extent of increasing resistance allele frequency around a hotspot of practical resistance takes time if bioassays are needed [50]. Ideally, a molecular assay can make a quick assessment possible, and the marker being developed by Farhan et al. [90] will be a welcome tool in this regard.

The same considerations of European corn borer movement ecology are important to consider when designing mitigation measures to slow the spread of resistance from a hotspot if it cannot be eradicated outright. Assuming resistant individuals behave similarly to susceptible adults, they will emigrate at a high rate from a hotspot of practical resistance, about 87% of them migrating and mating within a 12 km radius per generation. This is not necessarily a bad thing if the moths are migrating into areas of susceptibility, because most will emigrate before mating. The farther they migrate from the hotspot, the more likely they will encounter only susceptible adults with which to mate, producing only heterozygous susceptible offspring. Bioassays detected Cry1Ab resistance in a population collected in Kandiyohi, Minnesota in 2001 [56], and Cry1Fa resistance in a population collected in Hamilton County, Iowa in 2004 [60]. In both cases, subsequent collections and testing from the same sites showed no unusual survival, and it is likely that dilution of resistance alleles by emigration of resistant individuals and immigration of susceptible individuals to the hotspots of early warning resistance prevented further development in those areas. The importance of adequate refuge to produce susceptible moths is evident, as emphasized by Smith et al. [50] and Smith and Farhan [87], because of their provision of susceptible immigrants to mate with resistant residents in developing hotspots and a pool of susceptible moths to mate with resistant emigrants from hotspots.

Rates of resistance evolution to Bt-corn are likely to be greatest in regions where cornfields are relatively sparse and isolated in the larger landscape. The rate of emigration from the natal field is high, ~90% as proposed in this paper (Figure 4), based on multiple lines of evidence reviewed herein and in [89]. The question arises as to how resistance can evolve in a single field or cluster of fields when most of the natal population flies away before reproducing. When a migrant terminates its nonappetitive emigration flight, it may find itself fortuitously in or adjacent to an acceptable cornfield with acceptable grassy borders, in which case it will not have to do any searching for suitable habitat requiring more than station-keeping behavior. But if it lands in unacceptable habitat, it must begin ranging to locate a cornfield with associated grassy habitat. In a landscape with abundant, widespread cornfields, such as the U.S. Corn Belt, a searching moth will not have to range very far before its flight is arrested by suitable habitat. Thus, one expects few or none of the adult corn borers migrating out of the natal field will return to that field by ranging after migration. Conversely, in a landscape with only a few isolated cornfields, there is not the same expectation of no migrants returning to the natal field. The area around Wright’s neighborhood radius of ~12 km will harbor 87% of Bt-resistant emigrants from the isolated natal field of Bt-corn, some landing closer to their field of origin than others. When a moth emigrating from an isolated cornfield terminates migration, it will almost always be in non-corn habitat, and must engage in appetitive ranging behavior to find a cornfield. In a landscape with few cornfields, a ranging moth may travel a considerable distance before encountering one of the few acceptable fields available, with the effect that one or a few isolated cornfields serve to concentrate whatever European corn borer migrants have landed in the area. In such conditions, it is plausible that some Bt-resistant individuals will return fortuitously to their natal field to mate among themselves, the 10% of residents that never left in the first place, and immigrants (susceptible and/or resistant) mostly from the nearby landscape and some from many km distant that also colonize that field. Thus, the rate of positive assortative mating, which promotes resistance evolution [206], will be greater in a given field in the isolated landscape than in the Corn Belt. This scenario is hypothetical based on the logic of fundamental European corn borer movement behaviors, assumed to be the same regardless of landscape of origin. This concept could be tested in models that take into account a skewed distribution of emigrants around the natal field with 87% landing within 12 km (Figure 2), followed by ranging behavior among a few isolated cornfields (or isolated clusters of cornfields) in agricultural landscapes mimicking those common in Nova Scotia, the northeastern U.S., and western Great Plains.

Refuge function in IRM can be described in terms of source–sink dynamics, where refuges serve as sources of susceptible adults which colonize Bt crops. Each generation, the Bt crops are depopulated of larvae by the toxin(s) and thus act as a population sink [153,207]. As Bt resistance evolves, the Bt crop transitions from a sink to a source, leading to spread of resistance alleles. Ideally, the zone of influence of refuges (the area in which refuges increase the target insect’s population density [153]) can be determined and used to guide optimal distance of refuge placement from Bt fields, as Carrière et al. [153] illustrated for pink bollworm (*Pectinophora gossypiella*) in Bt-cotton. Deployment and rapid adoption of Bt-corn targeting European corn borer occurred several years before publication of this concept, and 0.8 km separation of refuge from a Bt field was settled on without a formal determination of refuge zone of influence. While this distance seemed reasonable based on information available in the late 1990s, a pulse of MRR [27,28,30,31,108,150,151] and population genetics (Table 1) studies were published in the early 2000s with a view to examine whether enough dispersal over 0.8 km really occurred to justify this criterion of refuge placement. Conclusions were unanimous that adequate dispersal did happen over that distance, but the authors also pointed out that actual dispersal distances were evidently much greater. Other evidence presented in this review supports this assessment.

Carrière et al. [208] pointed out that regional Bt resistance evolution and the source–sink dynamics of refuge function play out in metapopulations. The importance of this to European corn borer IRM is emphasized by implications of the conceptual framework of movement ecology proposed in this review (Figure 2 and Figure 4), including that the dimensions of metapopulations in areas like the Corn Belt are quite extensive. Predictions of this framework include that practical resistance to Bt-corn will develop soonest in landscapes of few cornfields, in part because the dimensions of the associated metapopulation are constrained by host availability and landscape, creating conditions of increased assortative mating between resistant insects emerging in those fields. In the Corn Belt, resistant migrants will spread resistance alleles through and beyond the local landscape, but will seldom mate with other resistant individuals emerging from the same field. Instead, frequency of resistance alleles are expected to increase slowly region-wide, while local instances of practical resistance and field failures will be rare and transient (see [56,60] for examples) until resistance allele frequencies eventually rise to the level of widespread early warning of resistance.

An important consideration is gene flow within and between the three partially reproductively isolated races of European corn borer. The boundaries of race distributions are not clearly demarcated. The BZ race is found throughout the species’ range in the U.S. and probably in the southern parts of Canada. The BE race is confined to roughly the eastern quarter of the species’ distribution in both countries [13,209]. The UZ race can be found throughout all of the species’ range in Canada and the northernmost states in the U.S. Hybridization is not rampant, but occurs regularly wherever there is overlap. Unless resistance genes are located on the sex chromosome where most of the reproductive isolating mechanism genes are located [43,46,47,48,49,210,211], resistance alleles in one race can be expected to introgress quickly into the other races through hybridization. The three races differ in other respects that might slow spread of resistance, including breadth of host plant utilization [13,209], although population genetics analyses do not support differentiated host races [44]. There is some indication that per-generation rates of dispersal may be higher in the BZ race than in the EZ and UZ races, based on rates of range expansion mentioned earlier. Evolution and spread within the bivoltine races is expected to be faster on a crop-year (multigeneration) timescale compared to the single-generation per crop-year UZ race. Regardless, movement distances and thus per-generation gene flow within all races are substantial (Table 1). Thus, resistance alleles are likely to spread rapidly, regardless of the race in which resistance initially evolves, especially once it reaches areas like the Corn Belt where the main host plant is ubiquitous and topographic barriers to movement are few.

In conclusion, I want to emphasize that remediation of field-evolved resistance of European corn borer to Bt-corn may not be a realistic option in most situations given the high mobility of adults and relatively long distances of per-generation gene flow. Containment of an isolated resistance hotspot may be possible, but the magnitude of pre-reproductive dispersal from the natal field (~90%) and the typical distances flown by the emigrants will necessitate rapid implementation of containment measures within an area surrounding the hotspot of at least 12 km radius [95], and probably greater [96]. If proposed mitigation measures include, especially, large-scale insecticide applications, the cost to the grower and to the environment must be weighed against the predicted efficacy of containment and long-term benefits of such a program. Such assessments will require modeling using realistic parameterization of moth behavior and gene flow. Current or enhanced IRM measures to slow resistance evolution should have some success in areas like the Corn Belt, where corn is the dominant component of the agricultural landscape and where increase in resistance allele frequency is probably proceeding region-wide. In all areas, the conditions that likely contributed to evolution of practical and early warning of resistance to Bt-corn detected in Canada [50,87] should be avoided wherever possible, such as planting single-toxin corn hybrids and the lack of structured refuge. Even in a landscape like the Corn Belt, the negative consequences of 5% RIB as a refuge strategy for a highly mobile insect like European corn borer, such as sublethal dosing of larvae [212,213], which can accelerate resistance evolution [8,68,214], will not likely be offset by an often assumed RIB advantage of increased mating of susceptible and resistant individuals emerging from the same field (e.g., [69]). While this assumption is applicable to a relatively sedentary insect like western corn rootworm [64,207,215], it does not apply to European corn borer, in which most emerging adults emigrate before mating and the area of genetic panmixia is ~450 km^2^ [27,86,89,95,205] (Figure 2 and Figure 4). In addition, compromised efficacy of Cry1Fa and other Cry proteins [50,87] endangers the justification for reducing the percentage refuge plants from 20% block refuge to 5% RIB for plants with pyramided toxins [85], made more acute by having reduced the source–sink ratio of refuge:Bt-corn in many landscapes from 20:80 to 5:95. Modeling these various dynamics in different kinds of landscapes using parameter values of dispersal components that reflect realistic adult European corn borer movement ecology is imperative in plotting the best path forward for delaying future and mitigating current practical resistance to Bt-corn targeting this pest.

## Figures and Tables

**Figure 1 insects-15-00160-f001:**
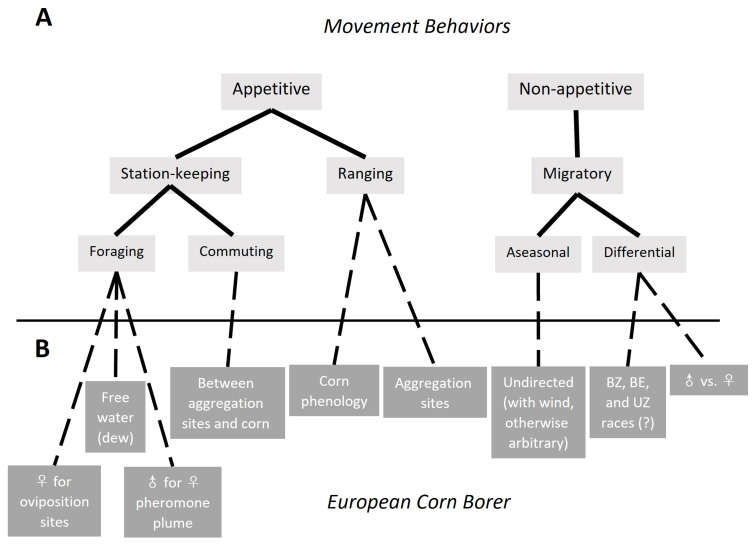
Types of movement behavior, as described by Dingle and Drake [100] and Dingle [101], which are exhibited by adult European corn borer. (**A**) Hierarchical categories of movement behavior, and (**B**) specific examples of such behaviors engaged in by European corn borer adults.

**Figure 2 insects-15-00160-f002:**
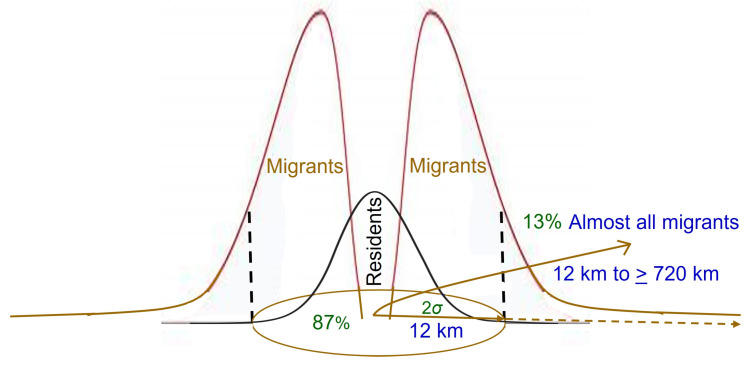
Schematic depiction of Wright’s genetic neighborhood area (4πσ^2^; foreshortened circle under dispersal distance frequency curves) of European corn borer calculated from population densities of adults sampled by flush bar at 68 sites in Marshall County, Iowa [26] and the isolation-by-distance slope generated with microsatellite genotype frequencies in populations from New York to Colorado as described by Kim et al. [95]. Approximately 87% of a population disperses a distance of 2σ per generation, which in the case of this species is ~12 km and is generated by the conflation of two dispersal kernels: that of the resident behavioral phenotype which disperses by appetitive station-keeping and ranging behaviors (black normal curve), and that of the migrant behavioral phenotype that disperses by nonappetitive migratory behavior (brown skewed curves with fat tails). The remainder of the population (~13%), presumably all or almost all of which are of the migrant behavioral phenotype, disperse beyond 12 km, up to 720 km according to pairwise *F*_ST_ estimates along transects in the central Corn Belt [33,94].

**Figure 3 insects-15-00160-f003:**
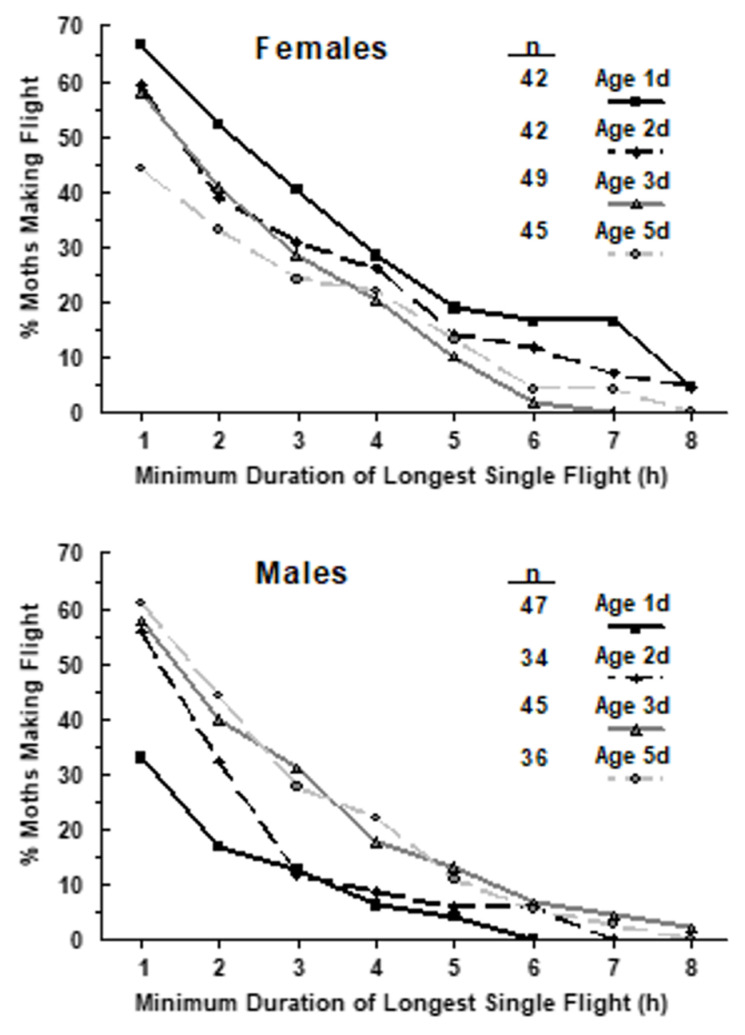
Decumulation curves showing the percentage of unmated European corn borer adults of different ages engaging in continuous, uninterrupted flights of indicated minimum durations on laboratory flight mills. Photoperiod in the flight chamber was 16L:8D, with only nocturnal flight behavior analyzed. No moths were tested the night of emergence. Age 1 indicates moths that emerged from the pupa the night before testing. From Dourhout et al. [97].

**Figure 4 insects-15-00160-f004:**
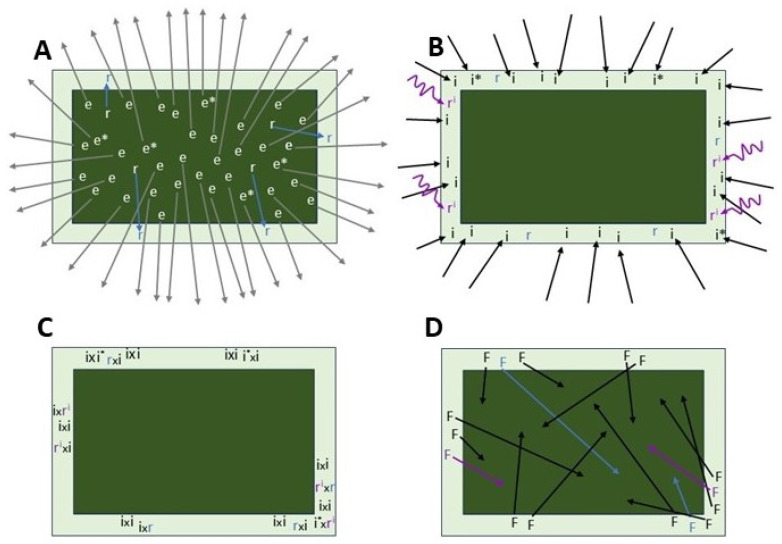
Conceptual model of adult European corn borer movement ecology relative to a subject cornfield (dark green) and its surrounding grassy borders (light green). This depiction assumes the crop is corn, the corn is at an attractive stage of maturity for oviposition, and the grassy borders are suitable for adult aggregation. (**A**) On a given night, roughly 90% of adults that emerge in the subject field are of the migrant behavioral phenotype (e), which emigrate beyond the field and its borders (gray arrows), most (~80%) before mating. Multiple directionality of arrows represents the undirected nature of European corn borer migration, but realized migratory flight direction on a given night will be strongly influenced by wind direction. About 13% of emergent moths will emigrate farther than 12 km (e*), the radius of Wright’s genetic neighborhood area estimated for European corn borer [95]. About 10% of emergent adults are of the resident behavioral phenotype (r), which leave the cornfield itself but settle in the adjacent grassy border (blue arrows) for daytime shelter and to mate the following night. (**B**) As most adults emigrate beyond the borders of their natal field, unmated immigrants (i) arrive from elsewhere (black arrows) and colonize the grass. Although the arrows for immigrants are depicted as straight lines, for most of these incoming moths the final approach to the grass after termination of migratory behavior will be by meandering ranging flight. About 13% of immigrants will have originated >12 km away (i*). A few of the immigrants may be of the resident behavioral phenotype (purple r^i^), which emigrated from their natal field in the local landscape because of inappropriate crop or grass conditions and encountered the target field by meandering ranging flight (purple wavy arrows). (**C**) Mating (x) between immigrants (i, i*), residents (blue r, purple r^i^), and each other occurs in aggregation sites in the grass borders, beginning the first or second night after emergence. (**D**) Mated females (F) enter the corn crop (arrows) on subsequent nights to oviposit. Black F indicates an immigrant of the migrant behavioral phenotype that earlier colonized the grass border of the target field to mate after emigrating by nonappetitive migratory flight from its own natal field located within or beyond the local landscape; purple F indicates an immigrant of the resident phenotype that earlier emigrated by appetitive ranging flight from its own natal field in the local landscape; and blue F indicates a resident ovipositing in its natal field. After ovipositing, many females return to the grassy border to spend the daylight hours ([26,27,28,108]), though not necessarily to the same aggregation sites.

## Data Availability

The data presented in this study are available in the article.

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
