# Peer review of "Critical Facets of European Corn Borer Adult Movement Ecology Relevant to Mitigating Field Resistance to Bt-Corn"

_insects, 2024, doi:10.3390/insects15030160_

Round 1

Reviewer 1 Report

Comments and Suggestions for Authors

This is an extremely needed and timely review of adult European corn borer (ECB) movement in light of the recent discovery of Bt resistance in ECB in Canada. This review is very well written and concisely summarizes decades of excellent research on the biology of ECB that has been forgotten due to the success of ECB management in North America using Bt corn.

This review also provides a revised understanding of previous assumptions on ECB movement with very important implications for Bt resistance mitigation. The author has vast experience in studying migratory insect behavior and the biology and management of European corn borer; therefore is extremely qualified to have completed this review.

 Minor suggestions to the text:

Line 43 – remove “other” before “crops”

Line 177 – I have not visited the northeastern US to experience the landscape around corn fields; however, I don’t know that Nova Scotia would be “even more” isolated than those. To be more objective, I would suggest this sentence be revised to read “Previous monitoring likewise had not been conducted in the Canadian Maritimes Provinces where relatively small cornfields may be isolated by forested areas and hills.”

Author Response

Comment: Line 43 – remove “other” before “crops”

Response: Done (L42).

Comment:  Line 177 – I have not visited the northeastern US to experience the landscape around corn fields; however, I don’t know that Nova Scotia would be “even more” isolated than those. To be more objective, I would suggest this sentence be revised to read “Previous monitoring likewise had not been conducted in the Canadian Maritimes Provinces where relatively small cornfields may be isolated by forested areas and hills.”

Response:  Done (L176-178).

Reviewer 2 Report

Comments and Suggestions for Authors

This is a well written document that summarizes much important information on Ostrinia.  It was very informative to read and will be a great contribution  to the debate regarding the situation in Canada at the moment.

The only reservation I have is the author’s suggestion that perhaps over 90% of ECB adults engage in migratory behavior the first night after emergence.  This suggestion seems to lay great weight on mark-release-recapture studies and low recapture numbers.  There are a multitude of problems with such studies, perhaps the most important is that there is no real control.  You don’t know, if no individuals dispersed, what proportion one might recover.  Traps relying on pheromones might be overwhelmed by high populations and large large local populations might also increase the likelihood of dispersal.  I have played around with numerous Fst simulators (genetic differentiation simulators) and if you set dispersal to 90% there is no local differentiation.  All fields in an area would essentially be genetically uniform, whether you simulate an island model or stepping stone model.  This also implies that resistance should evolve not in local patches but rather region wide.  Elevated resistance allele frequencies are present long before resistance is observed in bioassays and I would think that with such high gene flow rates resistance alleles would be would be present throughout a wide area.  Thus, this suggestion is not without repercussions. It would seem to make remediation difficult at best, and at worst requiring eradication of ECB over hundreds of kilometers.  

It occurs to me that another implication of the author’s suggestion that 90% of adults engage in migratory behavior is that a hypothetical isolated corn field would almost never have an ECB problem as 90% of the population would be lost each generation, no?  Refuges in particular would not be able to maintain populations due to the enormous sink effects of surrounding Bt-fields (see Carriere’s work).  

Ultimately I believe, that when making such a profound change to our understanding of the dispersal behavior of an insect, particularly a change that will drastically alter remediation plans, the burden of proof falls on the new proposal.  Unconventional reinterpretation of old data is unsatisfactory.  I would be much more likely to accept this interpretation of ECB dispersal behavior if it could be shown that local populations, perhaps all within 5km of each other, were virtually identical in gene frequencies.  That is the consequence of the current proposal that differs from our current understanding of ECB dispersal.  Unfortunately, to the best of my limited knowledge, those measurements have never been made.

I hasten to point out that I am not suggesting the author is incorrect.  The proposal is extremely intriguing, but I don’t think sufficient information is presented to support the idea.  More data, in my opinion, is needed.  Unfortunately, decisions regarding remediation need to made now and my concern is that the proposal in this manuscript will lead to enormous pesticide use in an attempt to eradicate ECB from large tracts of Canada.

Author Response

Comment:  The only reservation I have is the author’s suggestion that perhaps over 90% of ECB adults engage in migratory behavior the first night after emergence.  This suggestion seems to lay great weight on mark-release-recapture studies and low recapture numbers. 

Response: I made the mistake of relying too much on citing a detailed review of such data in a previous paper (Sappington 2018 [89]), which I have attempted to remedy in this revision. MRR results are not the only type of evidence for a high rate of emigration of European corn borer (ECB) from the natal field before reproduction, which also include flight mill studies, population genetics (including an estimate of Wright’s genetic neighborhood area radius), range expansion, moths captured far from the nearest likely source, and decoupled population densities in a field between generations. I have added quite a bit of information to the paper to address the need for more data. The added narrative, table, and figures should demonstrate the robustness of the interpretations in this review, or at least that they are not superficial or weakly supported. Detailed responses to the reviewer’s comments are below. 

Comment: There are a multitude of problems with such studies, perhaps the most important is that there is no real control. You don’t know, if no individuals dispersed, what proportion one might recover. 

Response: I have added a detailed section (3.2.2. Mark-Release-Recapture; Lines 401-470) on the eight MRR studies that have been conducted on this insect. The following paragraph (Lines 439-470) addresses the issues of knowing the number of individuals that dispersed from the release site and of what serves as a control:

“Just as a sudden appearance of large numbers of adults at a location can indicate an immigration event, a sudden disappearance from a location is evidence of emigration. Both kinds of events may go unnoticed if they occur via migratory behavior within a year-round distribution. While MRR studies cannot generally provide a clear picture of typical flight distances of released moths for the reasons outlined above, they can provide good evidence for mass departure from a release site when the experiment is designed to measure it. This is the case for most of the European corn borer MRR studies. Investigators in these studies generally assumed the released moths would occupy grassy field borders (or irrigated corn in arid regions) at or near the release site, but nearly all report very low mean percentages of nearby recovery, usually < 5% and often < 1%. The authors of all the European corn borer MRR studies concluded that most released moths must have flown beyond the furthest extent of recapture attempts based on evidence specific to their experimental set-ups, including spatial and temporal distributions of the recaptures, and comparisons with those of feral moths and their presumed source fields in the landscape [27,28,30,31]. In most studies, the number of moths that left the release site were either specifically known [1,27,28] or were calculated by subsampling adult emergence from corrugated cardboard rings of laboratory-reared internally-marked pupae [31,108,150,151]. Sampling in or near the release site was conducted in grass by the flush bar technique [150], light traps [30,31], pheromone traps [31], or sweep nets [1,27,28,108,151]. In the experiments reported by Dalecky et al. [27] and Bailey et al. [28], sweep netting was exhaustive in and near the grassy release sites, resulting in near 100% capture of any moths present. In all the MRR studies  (except [1]), the presence of feral moths in nearby sampled habitat provided a positive control for habitat suitability. That acceptable habitat near the release sites did not arrest flight activity of most released moths, a signature of nonappetitive behavior, suggests the departures were via migratory flight rather than via appetitive ranging. It is possible the high density of moths at a release site induced emigration by flight akin to escape behavior, which is appetitive. The releases in several studies  [30,31,102, 146] were of internally-marked pupae, from which adults were allowed to eclose in the field, thus reducing possible effects of handling on flight behavior. Although density-stimulated departure cannot be ruled out without additional experimentation, there are other lines of evidence supporting a high rate of emigration by adults from the natal field before reproduction.”

Comment: Traps relying on pheromones might be overwhelmed by high populations and large local populations might also increase the likelihood of dispersal. 

Response: I agree that pheromone traps are not the method of choice for recapturing marked moths, for a number of good reasons. However, only 1 of 8 MRR studies (the one by Showers et al. 2001 [150]) relied on pheromone traps alone for recaptures away from the vicinity of the release site. Even in that study, they used a flush-bar sampling method to detect moths in grassy areas in the immediate vicinity of the release sites to verify departure of the released moths. One other of the eight studies (Qureshi et al. 2005) used pheromone traps paired with light traps along transects. Light traps can also be problematic but less so than pheromone traps. Hunt et al. 2001 [30] relied solely on light traps. In the other 5 studies, recapture was by sweep net. I have acknowledged the weaknesses of using pheromone traps and light traps in the following text, Lines 422-438:

“Showers et al. [150] released a total of > 600,000 internally marked European corn borers the night after emergence over three years in six release-date windows of 3-5 separate releases each. Males were targeted for recapture by pheromone traps deployed in square rings out to a maximum distance of 75 km. Three marked females were recovered fortuitously during the course of the experiments. Male recapture rates ranged from 0.12-0.33% over the course of the study, most within 3.2 km, but trap density decreased with distance. Eleven marked moths were recaptured at distances > 7 km, with three recovered at > 40 km including a marked female “accidently” captured in a pheromone trap at 49 km from the release site. Use of pheromone traps or light traps for recovery in MRR studies of European corn borer [30,31,150] introduce another set of potential problems in interpretation of captures or non-captures, related to the traps’ reliance on successful manipulation of the insect’s behavior. Pheromone traps are especially problematic in that recovery data are restricted to males, the male must be sexually responsive during the trial, and capture efficiency may be affected by competition from feral females [154,155] and trap placement [16,156,157]. Nevertheless, maximum recapture distance represents the minimum distance the insect is capable of moving.”

Comment: I have played around with numerous Fst simulators (genetic differentiation simulators) and if you set dispersal to 90% there is no local differentiation.  All fields in an area would essentially be genetically uniform, whether you simulate an island model or stepping stone model.  This also implies that resistance should evolve not in local patches but rather region wide.  Elevated resistance allele frequencies are present long before resistance is observed in bioassays and I would think that with such high gene flow rates resistance alleles would be would be present throughout a wide area. 

Response: I agree to some extent, in that what the reviewer describes is probably what is happening curently in the Corn Belt. I have added a summary table of 13 published population genetics studies of European corn borer that report pairwise Fst’s and/or isolation-by-distance data using a variety of marker systems, and discuss them in a new section (3.2.4. Population Genetics and Gene Flow). These studies consistently show lack of genetic differentiation over large areas. But the issue that is somewhat hard to intuit at first is that while European corn borer movement behaviors are presumably the same everywhere, the location those behaviors ultimately take the insects to for reproduction (in terms of field colonization patterns) depends on the landscape they land in when migratory behavior terminates and ranging behavior begins. In other words, the resistance-evolution consequences of ECB flight behaviors will be different in the Corn Belt than in regions like Nova Scotia and the northeastern U.S. where fields of host plants are scattered, less common, and spatially delimited by features like forests and topography. The evolutionary consequences of ECB movement behavior depend on the landscape in which those behaviors are expressed. This concept and predictions for resistance evolution are explained now toward the end of the paper in Section 5 (Implications for Bt-Resistance Remediation and Mitigation), firstly at Lines 801-834:

“Rates of resistance evolution to Bt-corn are likely to be greatest in regions where cornfields are relatively sparse and isolated in the larger landscape. The rate of emigration from the natal field is high, ~90% as proposed in this paper (Figure 4), based on multiple lines of evidence reviewed herein and in [89]. The question arises as to how resistance can evolve in a single field or cluster of fields when most of the natal population flies away before reproducing, When a migrant terminates its nonappetitive emigration flight, it may find itself fortuitously in or adjacent to an acceptable cornfield with acceptable grassy borders, in which case it will not have to do any searching for suitable habitat requiring more than station-keeping behavior. But if it lands in unacceptable habitat, it must begin ranging to locate a cornfield with associated grassy habitat. In a landscape with abundant, widespread cornfields, such as the U.S. Corn Belt, a searching moth will not have to range very far before its flight is arrested by suitable habitat. Thus, one expects few or none of the adult corn borers migrating out of the natal field to return to that field by ranging after migration. Conversely, in a landscape with only a few isolated cornfields, there is not the same expectation of no migrants returning to the natal field. The area around Wright’s neighborhood radius of ~12 km will harbor 87% of Bt-resistant emigrants from the isolated natal field of Bt-corn, some landing closer to their field of origin than others. When a moth emigrating from an isolated cornfield terminates migration, it will almost always be in non-corn habitat, and must engage in appetitive ranging behavior to find a cornfield. In a landscape with few cornfields, a ranging moth may travel a considerable distance before encountering one of the few acceptable fields available, with the effect that one or a few isolated cornfields serve to concentrate whatever European corn borer migrants have landed in the area. In such conditions, it is plausible that some Bt-resistant individuals will return fortuitously to their natal field to mate among themselves, the 10% of residents that never left in the first place, and immigrants (susceptible and/or resistant) mostly from the nearby landscape and some from many km distant that also colonize that field. Thus, the rate of positive assortative mating, which promotes resistance evolution [206], will be greater in a given field in the isolated landscape than in the Corn Belt. This scenario is hypothetical based on the logic of fundamental European corn borer movement behaviors, assumed to be the same regardless of landscape of origin. This concept could be tested in models that take into account a skewed distribution of emigrants around the natal field with 87% landing within 12 km (Figure 2), followed by ranging behavior among a few isolated cornfields (or isolated clusters of cornfields) in agricultural landscapes mimicking those common in Nova Scotia, the northeastern U.S., and western Great Plains.”

And secondly after an intervening paragraph, Lines 852-866:

“Carrière et al. [208] pointed out that regional Bt resistance evolution and the source-sink dynamics of refuge function play out in metapopulations. The importance of this to European corn borer IRM is emphasized by implications of the conceptual framework of movement ecology proposed in this review (Figures 2 and 4), including that the dimensions of metapopulations in areas like the Corn Belt are quite extensive. Predictions of this framework include that practical resistance to Bt-corn will develop soonest in landscapes of few cornfields, in part because the dimensions of the associated metapopulation are constrained by host availability and landscape, creating conditions of increased assortative mating between resistant insects emerging in those fields. In the Corn Belt, resistant migrants will spread resistance alleles through and beyond the local landscape, but will seldom mate with other resistant individuals emerging from the same field. Instead, frequency of resistance alleles are expected to increase slowly region-wide, while local instances of practical resistance and field failures will be rare and transient (see [56,60] for examples) until resistance allele frequencies eventually rise to the level of widespread early warning of resistance.”

Comment: Thus, this suggestion is not without repercussions. It would seem to make remediation difficult at best, and at worst requiring eradication of ECB over hundreds of kilometers.

Response: I agree that remediation will rarely if ever be practical. A major implication of this paper should be to actually discourage eradication attempts, except possibly in ideal local, isolated conditions. Managers need to be aware that if remediation zones are not big enough, the ECB will blow right past the containment boundaries. In a region like the Corn Belt where topography does not help constrain the size of the metapopulation, the scale necessary to remediate would be so large that it should not even be attempted, because it is bound to fail. I have modified original text in various spots to avoid inadvertently suggesting remediation should be a goal under any except special conditions, and explicitly state this in a new paragraph at the end of the paper, Lines 888-920:

“In conclusion, I want to emphasize that remediation of field-evolved resistance of European corn borer to Bt-corn may not be a realistic option in most situations given the high mobility of adults and relatively long distances of per-generation gene flow. Containment of an isolated resistance hotspot may be possible, but the magnitude of pre-reproductive dispersal from the natal field (~90%) and the typical distances flown by the emigrants will necessitate rapid implementation of containment measures within an area surrounding the hotspot of at least 12-km radius [95], and probably greater [96]. If proposed mitigation measures include, especially, large-scale insecticide applications, the cost to the grower and to the environment must be weighed against the predicted efficacy of containment and long-term benefits of such a program. Such assessments will require modeling using realistic parameterization of moth behavior and gene flow. Current or enhanced IRM measures to slow resistance evolution should have some success in areas like the Corn Belt, where corn is the dominant component of the agricultural landscape and where increase in resistance allele frequency is probably proceeding region-wide. In all areas, the conditions that likely contributed to evolution of practical and early warning of resistance to Bt-corn detected in Canada [50,87] should be avoided wherever possible, such as planting single-toxin corn hybrids and the lack of structured refuge. Even in a landscape like the Corn Belt, the negative consequences of 5% RIB as a refuge strategy for a highly mobile insect like European corn borer, such as sublethal dosing of larvae [212,213] which can accelerate resistance evolution [8,68,214], will not likely be offset by an often assumed RIB advantage of increased mating of susceptible and resistant individuals emerging from the same field (e.g., [69]). While this assumption is applicable to a relatively sedentary insect like western corn rootworm [64,207,215], it does not apply to European corn borer, in which most emerging adults emigrate before mating and the area of genetic panmixia is ~450 km^2 [27,86,89,95,205] (Figures 2 and 4). In addition, compromised efficacy of Cry1Fa and other Cry proteins [50,87] endangers the justification for reducing the percentage refuge plants from 20% block refuge to 5% RIB for plants with pyramided toxins [85], made more acute by having reduced the source-sink ratio of refuge:Bt-corn in many landscapes from 20:80 to 5:95. Modeling these various dynamics in different kinds of landscapes using parameter values of dispersal components that reflect realistic adult European corn borer movement ecology are imperative in plotting the best path forward for delaying future and mitigating current practical resistance to Bt-corn targeting this pest.”

Comment:  It occurs to me that another implication of the author’s suggestion that 90% of adults engage in migratory behavior is that a hypothetical isolated corn field would almost never have an ECB problem as 90% of the population would be lost each generation, no? 

Response: No, this does not follow. First, please remember that even though migrant ECB can travel very long distances, most do not, ~87% landing within 12 km. In a landscape where cornfields are uncommon (i.e., isolated to varying degrees), those fields will concentrate migrants that have landed relatively nearby, including some ending up back in the natal field, because there will be few patches of acceptable habitat (cornfields of an appropriate phenological stage) to choose from. Please see the response above and the text in Section 5 at the end of the paper, Lines 801-834.

Comment: Refuges in particular would not be able to maintain populations due to the enormous sink effects of surrounding Bt-fields (see Carriere’s work).

Response: I have added a paragraph on the relation between refuge function and source-sink dynamics, Lines 835-851:.

“Refuge function in IRM can be described in terms of source-sink dynamics, where refuges serve as sources of susceptible adults which colonize Bt crops. Each generation, the Bt crops are depopulated of larvae by the toxin(s) and thus act as a population sink [153,207]. As Bt resistance evolves, the Bt crop transitions from a sink to a source, leading to spread of resistance alleles. Ideally, the zone of influence of refuges (the area in which refuges increase the target insect’s population density [153]) can be determined and used to guide optimal distance of refuge placement from Bt fields, as Carrière et al. [153] illustrated for pink bollworm (Pectinophora gossypiella) in Bt-cotton. Deployment and rapid adoption of Bt-corn targeting European corn borer occurred several years before publication of this concept, and 0.8 km maximum separation of structured refuge from a Bt field was settled on without a formal determination of refuge zone of influence. While this distance seemed reasonable based on information available in the late 1990s, a pulse of MRR [27,28,30,31,108,150,151] and population genetics (Table 1) studies were published with a view, in part, to examine whether enough dispersal over 0.8 km really occurred to justify this criterion of refuge placement. Conclusions were unanimous that adequate dispersal did happen over that distance, but the authors also pointed out that actual dispersal distances were evidently much greater. Other evidence presented in this review support this assessment.”

Response (continued): The enormous sink effect of Bt fields is the driving force behind areawide suppression of ECB that was achieved and has been maintained in the Corn Belt and other regions (Hutchison et al. 2010 [52], Dively et al. 2018 [10]), and populations remain very low in most areas. ECB is not an economic problem in refuges or unprotected fields, but they are not absent from the environment. I can, and do, go out every year here in Iowa and find at least a few ECB in grassy ditches near corn during both the spring and summer generations. It’s not like the old days before Bt-corn, and it takes a little longer to find them, but they’re out there. Pheromone traps catch a few males during the two generations we have here, and light traps catch even fewer of both sexes, so we can no longer obtain enough wild adults by trapping to initiate new lab colonies. Instead, targeting individuals in the grass for capture in sweep nets is how we obtain enough to initiate new laboratory colonies, which we do every second year or so. Alternative host plants may contribute to the overall populations in the eastern U.S., but not so much here in Iowa where commercial vegetable production is low. Not all farmers plant Lep targeting Bt-corn, and of course some can emerge from non-Bt plants in RIB fields. So the reviewer is correct in the sense that production of large numbers of ECB in refuge goes way down with high Bt adoption, but not to extinction. And the modeling of ECB population dynamics under the condition of refuge remaining in one spot year-to-year, under the unjustified assumption that susceptible populations will grow or be maintained spatially over time, is not applicable to this insect. The infestation in a field (or refuge) one generation is uncorrelated with the level of infestation in the preceding or the next generation (see new sub-subsection 3.2.5., Lines 551-576), which is why crop rotation or fall stalk destruction does not protect a particular field against ECB infestation. Instead, population densities (infestation levels) rise or fall across generations over much larger spatial areas than a single field or farm.

Comment:  Ultimately I believe, that when making such a profound change to our understanding of the dispersal behavior of an insect, particularly a change that will drastically alter remediation plans, the burden of proof falls on the new proposal.  Unconventional reinterpretation of old data is unsatisfactory.  I would be much more likely to accept this interpretation of ECB dispersal behavior if it could be shown that local populations, perhaps all within 5km of each other, were virtually identical in gene frequencies.  That is the consequence of the current proposal that differs from our current understanding of ECB dispersal.  Unfortunately, to the best of my limited knowledge, those measurements have never been made.

Response:  I have summarized Fst and isolation-by-distance results published in 13 population genetics studies of ECB from the U.S., France, Russia, and China in a new table (Table 1), along with a more thorough discussion of how such studies shed light on dispersal distances of ECB (Section 3.2.4). There are actually several population genetics studies reporting pairwise Fst’s over localized distances. I could find only one example at less than 5 km: a pair of locations separated by 2.7 km, where the Fst was 0.009 and nonsignificant (Malausa et al. 2007 [145]). However, there are 18 additional pairwise comparisons across three studies at < 16 km (4 of which were < 10 km); only one of these was significant (Fst = 0.013 at 12 km). In addition, temporal estimates of change in allele frequencies across 3-6 generations within a total of 10 sites across two independent studies were nonsignificant, showing stability of allele frequencies over time in the presence of gene flow from other non-differentiated sites in the region. I have added the following text to make these points (Lines 520-523):

“Pairwise comparisons of differentiation between nearby sites likewise indicated very low and nonsignificant Fst values. These included six comparisons between adult samples taken at 16-km intervals along an east-west transect in central Iowa [90]; three comparisons across distances of 7.5 to 13 km in Pennsylvania [42]; and 9 of 10 comparisons between sites in southern France separated by 2.7 to 16 km (one Fst of 0.013 at 12 km distance was significant) [145]. Most of the few significant Fst values reported for pairwise comparisons were either over long distances or were associated with areas of diverse cropping systems and fragmented agricultural landscapes. Bourguet et al. [172] noted that the only region in France with significant isolation by distance was in the northwest, where cornfields were fewer and more scattered than in the south and northeast regions of the country. Temporal estimates of change in allelic variation did not differ within locations at six sites over 4-6 generations in France [172], and temporal estimates of Fst were not significant within four sites over 3 generations in Iowa [94], indicating high stability of allele frequencies over time (Table 1).”

Comment:  I hasten to point out that I am not suggesting the author is incorrect.  The proposal is extremely intriguing, but I don’t think sufficient information is presented to support the idea.  More data, in my opinion, is needed.  Unfortunately, decisions regarding remediation need to made now and my concern is that the proposal in this manuscript will lead to enormous pesticide use in an attempt to eradicate ECB from large tracts of Canada.

Response:  In this revised version, I have added a large amount of material laying out the evidence for the conclusions informing the conceptual model of ECB movement ecology presented.here. To me, and with a healthy dose of trepidation (I’m too old to relish pushing back against well-entrenched paradigms, but it feels kind of necessary in this case given the stakes involved), the evidence from multiple angles (triangulated evidence so to speak) is quite clear that in a landscape like the Corn Belt, where reproductive habitat is omnipresent, a large proportion of ECB adults ovipositing in a cornfield came from somewhere else and mated nearby with others that also came from somewhere else; and that a reciprocally large proportion of ECB emerging in a field fly away before mating or ovipositing. Landscapes of relatively isolated fields like in eastern Canada and the northeastern US are different in that they serve to concentrate some of the ECB that have initially migrated a short distance away from their natal fields, because they aren’t going to run into many other cornfields as they search the landscape for suitable habitat. The evidence also indicates that not all ECB emerging in a field leave in the first place. While small in percentage terms (I’m proposing ~10%), these residents will have a much greater chance than in the Corn Belt of mating with each other or those that emigrated a relatively short distance and found their way back by ranging in a corn-depauperate landscape, which of course is the kind of assortative mating that can accelerate resistance evolution.

I share the concern of the reviewer that massive pesticide use to eradicate ECB from large areas is a bad idea and likely to fail  in most circumstances. I have added a new concluding paragraph to make this point clear (see earlier response, Lines 888-920). It is unlikely to succeed in part because of the high mobility of ECB adults, which has been known about in general for a century, but left out of models of resistance evolution I assume because the evidence for long-distance movement was confusing, scattered, obtained by different methodologies, deemed the exception rather than the rule, and had not been synthesized into a coherent whole. I attempted a thorough, holistic synthesis in an earlier publication [89] (in an obscure-ish journal, which doesn’t help, but it was an invited paper), and am attempting the same here in a slightly scaled-down version but within the specific context of Bt resistance. This review is not about reinterpreting old data per se, but synthesizing old and recent data, which together tell a consistent story. Reciprocally, the data compel abandonment of parts of the current story, because the data are inconsistent with some of the model assumptions about ECB IRM that are commonly used. Many of the assumptions about ECB adult movement used in models over the last quarter-century were never really supported by data in the first place, but were used because movement had to be parameterized in models “now” despite many unanswered questions and it seemed logical to use similar assumptions commonly used and justified for other insects in IRM models. This was understandable in the late 1990s and early 2000s when decisions had to be made about refuge size and distance based on limited data. But a bunch of studies have been conducted in the last 20+ years designed, in part, to test assumptions about refuge placed within 800 m (often designed to determine if ECB really do move that far, and interpreted by the original authors in that light), which indicated many of those initial assumptions were flawed - not in that 800 m was too far to ensure mixing, but to the various authors’ expressed-surprise that the ECB were apparently moving much farther than that, and so the bottomline conclusion at the time was yes, 800 m is not too far away. (See Lines 842-851.) All these educated guesses about ECB movement for parameterizing models were made in good faith. Yet without a synthesis of such studies to help modelers and regulators understand what ECB actually do movement-wise, and the lack of an imperative (i.e., no field resistance after 20 years, if it ain’t broke don’t fix it, relaxed interest in research and funding…), parameters of movement were not adjusted to keep pace with the knowledge gained. I do not think a more coherent, data-driven understanding of adult ECB movement ecology will lead to what the reviewer fears, and in fact should be a discouragement to such large-spatial-scale mitigation plans relying on conventional insecticides. Implementing an aggressive mitigation plan involving insecticides on a large enough spatial scale to have a chance of success based on the information in this review will be hard for anyone (politicians, farmers, regulators, general public, industry) to stomach for disruptive logistical and short-term economic reasons if nothing else. In contrast, under the current unrealistic assumptions about ECB adult movement being predominantly small-scale with mating and oviposition taking place within the natal field, it is easier to envision recommendations to nuke everything or prohibit corn planting etc. within a couple of km or so of a Bt field-failure. This kind of response not only would negatively impact the environment, especially in the aggregate, but economically as well because it would be a waste of effort as far as slowing Bt resistance evolution or spread. Given high adult mobility and their movement ecology, the best mitigation strategy (in addition to truly ending single-toxin products which is already understood and accepted by companies, although lagging in small-market areas like Nova Scotia that serve as a dumping ground for old products), would be to end the use of refuge-in-a-bag for this species. RIB increases sublethal doses in the larval stage while not counterbalancing it with increased mixing of susceptible and resistant ECB adults in a particular field, which the various types of data reviewed in this paper indicate simply does not happen because most of the adults leave the target field before mating. Banning RIB probably isn’t going to happen of course, because it would be hugely logistically inconvenient for companies and growers alike, despite its being counterproductive in slowing resistance evolution. I added a few comments about RIB in the last half of the concluding paragraph (Lines 904-916).

Reviewer 3 Report

Comments and Suggestions for Authors

The manuscript titled 'Critical Facets of European Corn Borer Adult Movement Ecology Relevant to Mitigating Field Resistance to Bt-corn' explores various aspects of European Corn Borer (ECB) adult movement and their implications for resistance management. Indeed, the manuscript contributes to the arena of Bt-corn resistance management by enhancing our understanding of ECB adult movement. However, the author mentioned in the conceptual framework that most ECB adults exhibit patterns of movement from aggregation areas to the host plant area (corn field) and then return to the aggregation area. Are there any driving factors for causing such patterned movement? This observation appears to contradict the statement on lines 220-221, which reads, 'Dispersal is not a behavior, but a spatial consequence of movement behavior.' Additionally, an illustration of the conceptual framework would be helpful in understanding ECB adult movement ecology.

Author Response

Comment:  the author mentioned in the conceptual framework that most ECB adults exhibit patterns of movement from aggregation areas to the host plant area (corn field) and then return to the aggregation area. Are there any driving factors for causing such patterned movement?

Response: The main driver seems to be high humidity, especially during daytime. Why females prefer to mate there at night is not understood, but probably is also related to humidity and free moisture in the form of dew. Most of what is known can be found in the references from Lines 53-61 (i.e., references [17-34]. Exploring the reasons for this behavior is beyond the scope of the paper. It is the fact that they commute that is important to understanding their overall movement ecology related to Bt resistance.  

Comment: This observation appears to contradict the statement on lines 220-221, which reads, 'Dispersal is not a behavior, but a spatial consequence of movement behavior.'

Response: It is not a contradiction. The movement behavior (which can result in dispersal) is a form of station-keeping called commuting. Experiments have shown that individual moths do not show spatial fidelity to a particular part of the grass (aggregation site) from one day to the next, and consequently they may oviposit in a different nearby field or different part of the same field on different nights. To help clarify, I have added a definition of dispersal as used in this review immediately preceding the line the reviewer references, Lines 222-223:

“Dispersal is defined in this paper as the geographic distance between parent and offspring [98].”

Comment:  Additionally, an illustration of the conceptual framework would be helpful in understanding ECB adult movement ecology.

Response: This is a very good suggestion. I have added a schematic figure with four panels (Figure 4) which illustrates the major points about ECB movement ecology relative to the natal field, which is the aspect most often misunderstood, and yet most important to IRM in Bt-corn. I also added a figure (Figure 2) to help the reader visualize the concept of Wright’s neighborhood area and radius as it applies to ECB as a partial migratory species.